# Photoantibacterial Poly(vinyl)chloride Films Applying Curcumin Derivatives as Bio-Based Plasticizers and Photosensitizers

**DOI:** 10.3390/molecules28052209

**Published:** 2023-02-27

**Authors:** Fábio M. S. Rodrigues, Iúri Tavares, Rafael T. Aroso, Lucas D. Dias, Carolina V. Domingos, Clara M. G. de Faria, Giusi Piccirillo, Teresa M. R. Maria, Rui M. B. Carrilho, Vanderlei S. Bagnato, Mário J. F. Calvete, Mariette M. Pereira

**Affiliations:** 1CQC Department of Chemistry, University of Coimbra, Rua Larga, 3004-535 Coimbra, Portugal; 2São Carlos Institute of Physics, University of São Paulo, São Carlos 13566-590, Brazil; 3Department of Biomedical Engineering, Texas A&M University, College Station, TX 77843, USA

**Keywords:** curcumin, PVC films, bioplasticizer, photosensitizer, antimicrobial photodynamic inactivation (aPDT), photosensitive material, photodecontamination

## Abstract

Herein we describe the design of natural curcumin ester and ether derivatives and their application as potential bioplasticizers, to prepare photosensitive phthalate-free PVC-based materials. The preparation of PVC-based films incorporating several loadings of newly synthesized curcumin derivatives along with their standard solid-state characterization is also described. Remarkably, the plasticizing effect of the curcumin derivatives in the PVC material was found to be similar to that observed in previous PVC–phthalate materials. Finally, studies applying these new materials in the photoinactivation of *S. aureus* planktonic cultures revealed a strong structure/activity correlation, with the photosensitive materials reaching up to 6 log CFU reduction at low irradiation intensities.

## 1. Introduction

One of the greatest challenges that polymer science faces today is the design and development of new bio-based polymer plasticizers [1,2] that can help minimize the environmental and public health damage resulting from the long lifetime of plastic materials [3] and also from the use of toxic phthalate-based plasticizers [4,5]. Indeed, phthalates are currently used as preferred additives in shaping the mechanical properties of many plastic materials, in particular those derived from polyvinylchloride (PVC), which have numerous end-uses such as building and construction, automotive, electrical and electronics, packaging, footwear, toys and healthcare devices [6]. Among the latter, we highlight the use of PVC–phthalate-based materials to manufacture life-support medical devices, such as endotracheal tubes [7]; these are often prone to developing multi-resistant bacteria and biofilms on their surface, consequently causing infections and, very likely, the patient’s death [8]. This has become a serious public health problem that has greatly contributed to the increasing number of deaths in patients with SARS-CoV-2, in addition to other pathologies that also require long-term intubation. In this regard, some of us [9] have recently reported the covalent linkage of curcumin to PVC–phthalate-based endotracheal tubes, which, after irradiation with blue light, have been shown to significantly decrease the formation of biofilms on their surfaces. Despite their extensive use, it is well established that, when released, phthalates may cause serious health damage [4,5]. For this reason, regulatory authorities in Europe and the United States are severely restricting their use [10,11]. In the recent literature, some studies have reported the use of bioplasticizers to shape the mechanical properties of PVC, aiming to, ideally, replace phthalates in the near future [2,12,13,14]. It has been demonstrated that the combination of curcumin, oxygen and light are key parameters for the inhibition of microorganisms by antimicrobial photodynamic inactivation (aPDI) [9,15,16,17,18,19,20,21,22,23,24].

Thus, we hypothesized that curcumin-based bioplasticizers could have optimized properties to simultaneously replace phthalates and inactivate bacteria by the action of light. Herein we evaluate the effect of the structure of curcumin ether and ester derivatives (including saturated fatty acids derived from waste oils) on the mechanical and photosensitive properties of PVC films, prepared with several loadings of the synthesized curcumin derivatives. All the films were characterized by standard means, including differential scanning calorimetry (DSC), thermogravimetry analysis (TGA) and mechanical tension. Finally, aPDI studies on the influence of curcumin derivatives films loading and light dose on *S. aureus* inhibition are presented and discussed.

## 2. Results and Discussion

### 2.1. Synthesis and Characterization of Curcumin-Based Photosensitive Plasticizers

The synthesis methodology for the preparation of the curcumin ester and ether derivatives is depicted in Figure 1a.

Regarding the preparation of curcumin ester derivatives, we implemented a strategic waste valorization, namely, using sunflower cooking oil (Figure 1b). First, to avoid the presence of reactive double C=C bonds in the hydrocarbon chains, we performed a Pd/C catalytic hydrogenation of cooking oil, followed by saponification [25]. The obtained mixture of saturated C14–C17 fatty acids was then activated by reaction with oxalyl chloride. Then, the corresponding acyl chloride formed in situ reacted with **1** in acetone, using triethylamine as base, at room temperature, over 24 h, yielding the curcumin fatty acid diesters **2** (C18) at 50% isolated yield. 

In order to evaluate the ester chain length in the plasticizing properties of the curcumin derivatives, the procedure was extended to the synthesis of curcumin diester C10 derivative **3**. This ester was prepared by reacting **1** with the commercially available decanoyl chloride in acetone, using triethylamine as base at 5 °C for 0.5 h. The compound was purified by flash chromatography yielding the desired curcumin derivative **3** in an 81% isolated yield. 

To further evaluate the influence of the type of alkyl chain, curcumin diether derivative **4** was prepared through the optimization of the nucleophilic substitution reaction between curcumin phenolic hydroxyl groups and *n*-decyl bromide (three equivalents) using 1,8-diazabicyclo [5.4.0]undec-7-ene (DBU) [26], as hindered base and DMF as aprotic solvent. After chromatographic purification, **4** was obtained in a 22% yield, along with the trisubstituted derivative, **5**, at 12%, Figure 1. All compounds were characterized using ^1^H-, ^13^C-NMR and IR spectroscopies and mass spectrometry (Appendix A)

Curcumin **1** and all the synthesized curcumin ether and ester derivatives **2**–**5** were characterized by UV-Vis spectroscopy using THF as solvent and the spectra are depicted in Figure 1 (see also Appendix A).

C10-diether derivative **4** shows a λ_max_ = 422 nm (Figure 1, red line), which is almost coincident with the typical curcumin (**1**) spectrum (Figure 1, black line) [27]. This indicates that the presence of electron-donating ether groups does not significantly change the electron conjugation. Contrarily, the sub-product C10-triether derivative **5** displays a λ_max_ = 380 nm. This large ipsochromic shift may be attributed to the reduction of conjugation across the molecule due to inhibition of the keto-enol equilibrium. Regarding the UV-Vis absorption spectra of diester curcumin derivatives **2** and **3**, shorter ipsochromic shifts are observed, λ_max_= 400 nm (Figure 1 violet, blue and yellow lines), when compared with **5** [28,29]. This occurrence may be attributed to the reduction of conjugation due to the transformation of curcumin hydroxyl groups to the correspondent electron-withdrawing ester groups.

Table 1 shows the values of the absorption maxima and molar absorptivity (ε) for all the synthesized compounds, where the triether substituted compound **5** shows the lowest molar absorptivity value (26,665 M^−1^ cm^−1^). Thus, considering that the synthesis of ideal photosensitizers and plasticizers derived from curcumin is the main goal of this work, compound **5** was not further studied due to its low synthetic yield and less suitable visible light absorption.

As mentioned earlier, the efficiency of photodynamic therapeutics relies largely on the ability of the photosensitizer to generate singlet oxygen (^1^O_2_), whose quantum yield (Φ_Δ_) depends on the O_2_ concentration, the nature of the solvent and the photosensitizer’s triplet state lifetime. Therefore, we calculated the singlet oxygen quantum yields of compounds **1**–**4** by the time-resolved phosphorescence emission technique, using phenalenone as standard and THF as solvent [30], and the results are presented in Table 1. Once again, the ether derivative **4** shows close similarities to curcumin **1** (Φ_Δ_(**1**) = 0.372 and Φ_Δ_(**4**) = 0.333) while the ester derivatives (**2** and **3**) presented slightly lower singlet oxygen quantum yields. Concerning fluorescence quantum yields, a comparative method was also used with quinine sulfate as standard (Φ_F_ = 0.546 [31]) and the results are shown in Table 1. For curcumin Φ_F_(**1**) = 0.101 was obtained, which is in agreement with the literature using the absolute method (curcumin Φ_F_ = 0.14) [32]. From the analysis of Table 1, we observe that the esterified derivatives **2** and **3** show 10-fold lower Φ_F_ then curcumin **1** or its ether derivative **4**.

### 2.2. Photodegradation Evaluation of Curcumin-Based Plasticizers

In order to evaluate the photostability of curcumin **1,** curcumin derivatives **3** and **4** as photosensitizers and plasticizers, their photobleaching profiles were determined after irradiation (at 450 nm) using different light doses (0, 9.4 and 23.5 J/cm^2^) (Figure 2). Photostability percentages were obtained by analyzing the decrease of the maximum absorbance peaks. For all groups, an increase in photodegradation was observed with increasing light dose showing the ability of photosensitizers to generate oxidative species with increasing dose, while showing instability towards photooxidation. For curcumin-based plasticizers **3** and **4**, a degradation between 70 and 88% was observed using a light dose of 9.6 J/cm^2^. Among them, the ether-type derivative showed a lower photodegradation than its ester-type analog, which demonstrates the relevance of the curcumin hydroxyl protection functional group in the ether form.

### 2.3. Preparation and Characterization of PVC–Curcumin-Based Films

To evaluate the potentialities of curcumin (**1**) and the curcumin derivatives (**2**–**4**), both as photosensitizers and bioplasticizers, different PVC–curcumin-based films were prepared, using a doctor blade type film coating procedure (Appendix A) [33]. To evaluate the effect of the amount of curcumin derivative on the film properties, ~25 µm thick films with several loads (0.1%, 15% and 30% *w*/*w* curcumin derivative: PVC) were prepared (Figure 3).

To prevent UV-Vis spectral saturation, films with just 0.1% (*w*/*w* curcumin derivative/PVC) were used to measure the corresponding UV-Vis and results are presented in Figure 4. Notably, the incorporation of curcumin (**1**), as well as curcumin-based ether and ester derivatives (**2**–**4**) on the phthalate-free PVC did not show significant changes on their light absorption properties when compared with the absorption of the compounds in solution (Figure 1). This is an important property regarding their use as potential photosensitive materials for the preparation of antimicrobial surfaces.

### 2.4. Thermal Stability and Mechanical Properties

To evaluate their potential as PVC-based material, the thermal and stress–strain properties of these films were further studied. Since the thermostability is an important property of PVC films, thermogravimetric analysis (TGA) was performed for the phthalate-free PVC and the PVC–curcumin-based films PVC(**1**)-curc, PVC(**2**)-esterC18, PVC(**3**)-esterC10 and PVC(**4**)-etherC10 (30% *w*/*w* curcumin derivative/PVC) in the temperature range 25–800 °C, at a heating rate of 10 °C/min under 20 mL/min nitrogen purge; the TGA plots are depicted in Figure 5.

The thermal decomposition of PVC, performed in N_2_ atmosphere, was shown to occur into three stages: the first is in the temperature range 50 to 150 °C, and relates to evaporation of polymer-coordinated solvent molecules; the second stage, between 200 and 400 °C is attributed to the elimination of Cl atoms, in the form of HCl, from the PVC; the third stage begins at 400 °C and ends at 560 °C, in which thermal degradation of the PVC chains occurs, producing volatile compounds by intramolecular cyclization of the conjugated chains in agreement with the literature [14]. This phenomenon occurs similarly for the PVC–curcumin film (see Appendix A), in which the addition of curcumin to PVC does not affect the thermal decomposition profile when compared to PVC (black), which demonstrates the high thermal stability of the doped PVC–curcumin material. The same thermal behavior was observed for all the other PVC–curcuminoid films.

Another crucial property to evaluate the potentialities of these new curcumin ether and ester derivatives as PVC bioplasticizers is their glass transition temperature (Tg), measured by differential scanning calorimetry (DSC), which outlines a borderline region between mainly glassy and highly elastic states [34]. First, the films containing 15% *w*/*w* curcumin derivative/PVC were studied. However, independently of the curcumin derivative used, no significant changes in Tg were observed which demonstrates that this curcumin loading is scarce. Therefore, the films containing 30% *w*/*w* curcumin derivative/PVC were selected to pursue this study and the glass transition temperatures (Tg) of pure and PVC–curcumin films are presented in Table 2 (the corresponding DSC curves are shown in Appendix A).

The phthalate-free PVC presents a Tg = 73 °C (Table 2, entry 1), which agrees with the literature (75 °C) [35]. From the analysis of the Tg values shown in Table 2, a significant effect of the curcumin structure on the plasticization properties can be observed. In fact, the doping of PVC with 30% curcumin caused only an 8 °C decrease in the Tg value and the doping with a long-chain ester (C18) even caused an increase in the Tg value. (Table 2, entries 2 and 3). On the other hand, the reduction of the chain size of the ester derivative (C10) caused a significant decrease of the Tg value (ca −18 °C) (Table 2, entry 4). Replacing the ester with an ether group with the same C10 chain size leads to a pronounced effect on the transition glass temperature obtaining a significantly lower Tg = 55 °C (Table 2, entry 5). 

Another relevant property for assessing the potential of a new molecule as a plasticizer is the mechanical tensile strength. Thus, the plasticizing efficiencies of the new curcumin diester and diether derivatives were evaluated by performing tensile tests. The PVC-based films containing curcumin derivatives PVC(**1**)-curc, PVC(**2**)-esterC18, PVC(**3**)-esterC10 and PVC(**4**)-etherC10, with a 30% *w*/*w* ratio of curcumin derivative to PVC, were used to perform the studies of mechanical stress. The values obtained for tension at maximum strength (σ_M_) and elongation at breaking tension (ε_tB_) are presented in Table 2. Comparison of the elongation and breaking results between pure PVC and the derivatives doped with curcumin and its esters did not show a very significant effect. On the contrary, the film prepared with PVC and curcumin-diether C10 **4** (30% *w*/*w*) produced a significant decrease in the tensile strength, and an increase in elongation at breaking tension (σ_M,_ 18.36 ± 2.33 and ε_tB_ 111.00 ± 3.3; Table 2 entry 5). As selected examples, Figure 6 shows the comparative stress–strain curves for phthalate-free PVC films and plasticized PVC(**4**)-etherC10 films (see also Appendix A). The clearly improved elongation at break in PVC(**4**)-etherC10 can be attributed to the greater interlocking between PVC chains after the addition of curcumin ether C10 plasticizing molecules. The rigidity of PVC is softened by these moderate chain molecules, by separating the long PVC chains. In addition, the increase in the free volume between PVC molecules caused by the incorporation of bioplasticizer (**4**) significantly reduces the modulus at the breaking strength. These results are in agreement with previously described plasticizing effects for other bioplasticizers [36,37].

Furthermore, the mechanical studies correlate well with thermogravimetric analysis, with a glass transition being observed for the PVC(**4**)-ether C10 material (Tg = 55 °C, entry 3, Table 1), which is about 20 °C below the Tg of PVC [38,39], being even more pronounced than that observed for phthalate-containing PVC. Therefore, we can conclude that curcumin-ether derivative (**4**) is suitable for the purpose of being incorporated as a bioplasticizer in PVC polymer. Regarding the curcumin ester derivatives (**2**) and (**3**), the polarized interaction of the carbonyl group with PVC and the increased molecular weight of the compounds decrease their ability as plasticizer.

### 2.5. Antibacterial Photoinactivation Studies

To demonstrate the potential of these films as photosensitive antibacterial materials, *S. aureus* cultures were used as a model. Circular sections of the PVC films were placed in 96-well plates and *S. aureus* planktonic cultures were added. Then, aqueous bacterial suspensions were placed above the films and immediately irradiated with a blue LED lamp (415 nm), using different light doses. Finally, the number of surviving colony-forming units (CFU) after photoinactivation treatment were counted. In Figure 7a, a comparative study is shown for the PVC–curcumin films, PVC(**1**)-curc, with different curcumin loads (5–30% *w*/*w*). The materials did not show any effectiveness in the dark, which shows that light-promoted ROS generation is the main mechanism associated with the observed antimicrobial activity. At the lowest light dose (9.4 J/cm^2^), a relatively small bactericidal effect (<1 log CFU) was observed in neat PVC, compared with the control. The photosensitive material PVC(**1**)-curc with 5% and 15% *w*/*w* curcumin loads did not show any improvement in bactericidal activity when compared with PVC; however, an additional 1.5 log CFU *S. aureus* reduction was obtained with PVC(**1**)-curc 30% *w*/*w*. The photoinactivation was light-dose-dependent, since at 23.5 J/cm^2^, a significant anti-*S. aureus* activity (4 log CFU reduction) was achieved with PVC(**1**)-curc 5–30% *w*/*w*. Since the 30% *w*/*w* PVC–curcumin loads showed a better overall antimicrobial activity, a comparative study for all the PVC–curcumin-based materials, PVC(**1**)-curc, PVC(**2**)-esterC18, PVC(**3**)-esterC10 and PVC(**4**)-etherC10 with 30% *w*/*w* loads was carried out (Figure 7b).

At 9.4 J/cm^2^, no significant difference in photodynamic efficiency was obtained among PVC(**1**)–(**4**) (0 to 2 log CFU reduction compared with neat PVC). However, at 23.5 J/cm^2^, a significant photoinactivation was achieved with PVC(**3**)-esterC10 and PVC(**4**)-etherC10, both exhibiting 6 log CFU reduction. This antimicrobial activity complies with the guidelines of the US Environmental Protection Agency for healthcare facility disinfection, which require ≥ 6 log CFU reduction to substantiate an efficient disinfection claim [40]. Interestingly, these materials showed an additional 3 log CFU reduction when compared to PVC(**1**)-curc. This may be due to the improved amphiphilic properties of the PVC–curcumin derivatives PVC(**3**)-esterC10 and PVC(**4**)-etherC10, both having C10 alkyl chains, which might promote a better approximation of *S. aureus* cell walls to the photosensitive material. It is well known that singlet oxygen, one of the main ROS produced by curcumin [22], has a relatively short lifetime (τ_∆_ ≈ 3 µs), which allows a diffusion radius of only ≈200 nm in aqueous solutions [41]. Thus, it is expected that a closer contact between bacteria and PS may greatly improve the inactivation efficiency of the generated ROS.

### 2.6. Cytotoxicity Evaluation

According to physical/chemical analysis and the antibacterial photoinactivation studies, we have selected PVC(**4**)-etherC10 for cytotoxicity studies, since it showed the most promising properties to be applied both as a photosensitizer and plasticizer. In this regard, Figure 8 shows in vitro cytotoxicity evaluation of curcumin-based plasticizer **4**. These experiments were performed using human fibroblast cell lines (HDFn) under different concentrations (at 2.5, 5, and 10 µg/mL). It should be noted that no significant difference is found when comparing the photosensitizer’s cytotoxicity to their equivalent formulation control (with the same solvent concentrations) at all concentrations tested (at 2.5, 5 and 10 µg/mL).

## 3. Materials and Methods

### 3.1. Materials and Methods

All solvents and chemicals were purchased from Sigma Aldrich, TCI Europe and Alfa Aesar and used as received. Particularly, PVC (average Mw ~80,000, average Mw ~47,000) was Sigma Aldrich product no. 389323 (CAS no. 9002-86-2, St. Louis, MO, USA). Curcumin (**1**) was synthesized according to the literature [42]. A sample of hydrogenated used cooking oil containing a mixture of glycerol triesters was prepared according to the literature [43]. Characterization data of compound **2** are in agreement with the literature [12]. Nuclear magnetic resonance (NMR) spectra were obtained using a Bruker AVANCE 400 MHz spectrometer. Tetramethylsilane (TMS) was used as reference and deuterated chloroform was used as solvent. Mass spectra were obtained using a Bruker Microtof spectrometer (Bruker, Billerica, MA, USA) equipped with a selective ESI detector, belonging to the Mass and Proteomics Unit of the University of Santiago de Compostela, Spain. UV-Vis spectra were recorded on a Hitachi U-2010 spectrophotometer, using quartz cells with an optical path of 1 cm or an adapter for solids. Melting points were determined on an Electrothermal-Melting Point Apparatus capillary microscope. Fourier-transform infrared spectroscopy (FTIR) was performed using a ThermoNicolet IR380 apparatus.

### 3.2. Synthesis of Curcumin-Ester-Based Plasticizers

Synthesis of curcumin fatty acid ester derivative (**2**)

The synthesis of curcumin fatty acid ester derivative (**2**) was performed in three steps, described below.

Pd/C catalyzed hydrogenation of cooking oil unsaturated triglycerides*:*. Cooking oil (5.0 mL) and 5% Pd/C catalyst (150 mg) were introduced into a stainless-steel autoclave. The reactor was then pressurized with hydrogen (P = 10 bar) and kept, under stirring, for 1 h at 80 °C. After cooling to room temperature, the autoclave was slowly depressurized, the crude result was dissolved in chloroform and filtered to remove the Pd/C catalyst. Finally, after solvent evaporation and drying under vacuum, the saturated triglyceride mixture was obtained as a white solid (ca. 4.0 g).

Triglyceride saponification: A beaker containing a mixture of hydrogenated triglycerides (1.5 g) was placed on a stirring plate, with heating at 70 °C. Then, a 1M KOH solution (10 mL), ethanol 95% (50 mL) and distilled water (10 mL) were added. The mixture was left under stirring for 2 h. After this time, a decantation was made to remove any residues. Then, a 6M HCl solution (100 mL) was added, under stirring in an ice bath, until it reached pH = 1. The precipitate was then filtered into a 100 mL porous plate funnel and the solid was washed with hexane (3 × 50 mL). Finally, the solid was dissolved in dichloromethane and dried with anhydrous sodium sulfate. After filtration, solvent evaporation and drying under vacuum, the fatty acid mixture (mostly stearic) was obtained (1.1 g), in agreement with the literature [25].

Curcumin esterification: The fatty acids mixture previously prepared (3.16 × 10^−3^ mol, 0.9 g) was placed in a 100 mL two-neck round bottom flask under inert atmosphere in an ice bath. Dry dichloromethane (15 mL) and dry THF (5 mL) were added to dissolve the acid, and finally 5 drops of DMF were added. Under stirring, oxalyl chloride (4.22 × 10^−3^ mol, 0.36 mL) was added dropwise and the reaction was left for 1.5 h at about 5 °C, then 30 min at room temperature. In parallel, in a 100 mL flask, a solution of curcumin (7.91 × 10^−4^ mol, 0.29 g), 10 mL of dry THF and triethylamine (7.91 × 10^−3^ mol, 1.1 mL) was prepared. Using a syringe, the fatty acid chloride solution was transferred and added dropwise to the curcumin solution via septum. The reaction was then left stirring for 24 h, under an inert atmosphere. After this point, 30 mL of distilled water and 45 mL of dichloromethane were added and the reaction mixture was extracted with 1M sodium bicarbonate solution (4×) and with brine (2×). The reaction crude product was then dried with anhydrous sodium sulfate followed by a chromatographic column on silica gel, using a 4:1 mixture of *n*-hexane:ethyl acetate as eluent to separate unreacted acid and other impurities. Next, crude product was eluted with dichloromethane and precipitated by addition of *n*-hexane. After filtration, washing with *n*-hexane (200 mL), and drying under vacuum, the curcumin ester **2** was isolated with a 50% yield (0.36 g).

^1^H NMR (400 MHz; CDCl_3_): δH, ppm: 7.62 (d, *J* = 16.0 Hz, 2H), 7.16 (d, *J* = 8.0 Hz, 2H), 7.12 (s, 2H), 7.05 (d, *J* = 8.0 Hz, 2H), 6.56 (d, *J* = 16.0 Hz, 2H), 5.86 (s, 1H), 3.87 (s, 6H), 2.58 (t, *J* = 6.4 Hz, 4H), 1.78 (m, 4H), 1.42 (m, 4H), 1.26 (m, ~52H), 0.88 (t, *J* = 8.0 Hz, 6H); ^13^C NMR (100 MHz, CDCl_3_) δC, ppm: 183.3; 171.8, 151.6, 141.6, 140.2, 134.0, 124.3, 123.5, 121.3, 111.6, 101.9, 56.1, 34.2, 32.1, 29.8, 29.8, 29.8, 29.7, 29.5, 29.4, 29.2, 25.2, 22.8, 14.3; ESI-MS (m/z): calcd for C_57_H_89_O_8_ = 901.6557, found for [M+H]^+^ = 901.4171; UV-Vis (λ_max_) in THF: 400 nm, Ɛ: 49,093 M^−1^ cm^−1^; M. P.: 75–79 °C; FTIR: 2916 cm^−1^ (ν OH), 2849 cm^−1^ (ν CH), 1740 cm^−1^ (ν C=O) and 1119 cm^−1^ (ν C-OR).

#### Synthesis of Curcumin-decanoyl Ester Derivative **3**

To a solution of curcumin (1 g, 2.71 × 10^−3^ mol) and NEt_3_ (1.1 mL, 8.14 × 10^−3^ mol) in dry acetone (10 mL), decanoyl chloride (5.5 × 10^−3^ mol) was added at 0–5 °C. The mixture was stirred at this temperature for 0.5 h. Then, the mixture was quenched with water (30 mL) and extracted with dichloromethane (50 mL). The combined organic layer was washed with brine (3 × 30 mL), water (2 × 30 mL) and dried over anhydrous Na_2_SO_4_. The crude was then concentrated under reduced pressure and the formed solid was purified by flash chromatography using dichloromethane as eluent. Compound **3** was obtained at an 81% yield (1.23 g).

((1*E*,3*Z*,6*E*)-3-hydroxy-5-oxohepta-1,3,6-triene-1,7-diyl)bis(2-methoxy-4,1-phenylene) bis(decanoate) (**3**)

^1^H NMR (400 MHz; CDCl_3_): δH, ppm: 7.62 (d, *J* = 15.8 Hz, 2H), 7.16 (d, *J* = 8.2 Hz, 2H), 7.12 (s, 2H), 7.05 (d, *J* = 8.2 Hz, 2H), 6.56 (d, *J* = 15.8 Hz, 2H), 5.83 (s, 1H), 3.87 (s, 6H), 2.59 (t, *J* = 7.5 Hz, 4H), 1.77 (m, 4H), 1.42 (m, 4H), 1.28 (m, 20H), 0.89 (t, *J* = 6.8 Hz, 6H); ^13^C NMR (100 MHz, CDCl_3_) δC, ppm: 183.3, 171.8, 151.6, 141.6, 140.2, 134.0, 124.3, 123.5, 121.3, 111.6, 101.9, 56.1, 34.2, 32.0, 29.4, 25.2, 22.8, 14.3; ESI-MS (m/z): calcd for C_41_H_56_O_8_Na = 699.3873, found for [M + Na]^+^ = 699.3856; UV-Vis (λ_max_) in THF: 400 nm, Ɛ: 43,682 M^−1^ cm^−1^; M. P.: 84–88 °C; FTIR: 2911 cm^−1^ (ν OH), 2844 cm^−1^ (ν CH), 1766 cm^−1^ (ν C=O) and 1115 cm^−1^ (ν C-OR).

### 3.3. Synthesis of Curcumin-Ether-Based Plasticizers

Curcumin (0.1 g, 2.7 × 10^−4^ mol) was placed in a 50 mL round bottom flask and DMF (1 mL) was added. Then, DBU (0.08 mL, 5.4 × 10^−4^ mol) and bromodecane (0.17 mL, 8.1 × 10^−4^ mol)) were consecutively added to the flask. The reaction mixture was stirred at room temperature for 20 h (followed by TLC). Upon completion, dichloromethane was added (100 mL) and the crude product was washed with distilled water (7 × 50 mL) and dried over Na_2_SO_4_. Solvents were evaporated and the crude product was subjected to flash chromatography using a 4:1 mixture of *n*-hexane:ethyl acetate. The first eluted minor fraction was the trisubstituted compound, while the second (main fraction) was the desired disubstituted compound. Compound **4** was obtained with 22% yield (0.04 g) and compound **5** was obtained with 12% yield (0.03 g).

(1*E*,4*Z*,6*E*)-1,7-bis(4-(decyloxy)-3-methoxyphenyl)-5-hydroxyhepta-1,4,6-trien-3-one (bis-decyloxy-curcumin) (**4**)

^1^H NMR (400 MHz; CDCl_3_): δH, ppm: 7.60 (d, *J* = 15.8 Hz, 2H), 7.12 (d, *J* = 8.6 Hz, 2H), 7.08 (s, 2H), 6.87 (d, *J* = 8.6 Hz, 2H), 6.49 (d, *J* = 15.8, 2H), 5.82 (s, 1H), 4.05 (t, *J* = 6.88 Hz, 4H), 3.91 (s, 6H), 1.86 (m, 4H), 1.46 (m, 4H), 1.27 (m, 24H), 0.88 (t, *J* = 6.74 Hz, 6H); ^13^C NMR (100 MHz, CDCl_3_) δC, ppm: 183.4, 150.9, 149.7, 140.6, 128.0, 122.8, 122.0, 112.6, 110.4, 101.4, 69.2, 56.2, 32.0, 29.7, 29.4, 29.3, 29.0, 26.1, 22.8, 14.3; ESI-MS (m/z): calcd for C_41_H_61_O_6_ = 649.4463, found for [M + H]^+^ = 649.4456; UV-Vis (λ_max_) in THF: 421 nm, Ɛ: 42,695 M^−1^ cm^−1^; M. P.: 74–77 °C; FTIR: 2921 cm^−1^ (ν OH), 2845 cm^−1^ (ν CH).

(1*E*,4*Z*,6*E*)-5-(decyloxy)-1,7-bis(4-(decyloxy)-3-methoxyphenyl)hepta-1,4,6-trien-3-one (tris-decyloxy-curcumin (5)

^1^H NMR (400 MHz; CDCl_3_): δH, ppm: 8.20 (d, *J* = 16.0 Hz, 1H); 7.56 (d, *J* = 8.0 Hz, 1H); 7.35 (d, *J* = 8.0 Hz, 1H); 7.20–6.80 (m, 6H); 6.74 (d, *J* = 16.0 Hz, 1H); 5.71 (s, 1H); 4.04 (m, 2H); 3.99 (t, *J* = 8.0 Hz, 4H); 3.93 (s, 3H); 3.91 (s, 3H); 1.86 (m, 6H); 1.44 (m, 6H); 1.27 (m, 36H); 0.89 (t, *J* = 8.0 Hz, 9H); ^13^C NMR (100 MHz, CDCl_3_) δC, ppm: 193.2, 192.8, 183.4, 149.7, 140.6, 137.5, 131.5, 128.0, 122.8, 122.0, 119.1, 112.8, 112.6, 110.4, 110.3, 101.4, 69.2, 63.3, 56.2, 33.0, 32.0, 29.7, 29.5, 26.1, 22.8, 14.3; ESI-MS (m/z): calcd for C_51_H_80_O_6_ = 787.5955, found for [M]^+^ = 787.5866; UV-Vis (λ_max_) in THF: 380 nm, Ɛ = 26665 M^−1^ cm^−1^; M. P.: 57–60 °C; FTIR: 2926 cm^−1^ (ν OH), 2844 cm^−1^ (ν CH).

### 3.4. Photophysical and Photochemical Characterization of Curcumin-Based Plasticizers

Oxygen singlet quantum yields were determined by comparing the initial emission intensity of the optically equilibrated study solutions with a phenalenone standard solution in THF. The wavelength of 1270 nm was selected for detection on the Hamamatsu R5509-4 photomultiplier, cooled to 193 K in a liquid nitrogen chamber. A solution of phenalenone in THF, φ_Δ_ = 0.96 was used as a standard [30]. Steady-state fluorescence studies were performed on a Fluoromax^®^ series spectrofluorimeter (HORIBA Scientific) using four-sided cells (Hellma) with an optical path of 1 cm. Fluorescence spectra, increments and times of 1 nm per 1 s of integration were considered. For the fluorescence emission spectra, the excitation was performed at the wavelength of maximum absorption (350 nm), and fluorescence spectra were obtained in the range of 360 nm to 690 nm. All fluorescence excitation spectra were corrected for the instrumental response of the system used. A solution of quinine sulfate in in H_2_SO_4_ 0.1 M (Φ_F_ = 0.546) was used as a standard [31].

### 3.5. Photodegradation Evaluation of Curcumin-Based Plasticizers

Photodegradation evaluation was performed using a light-source device (Biotable^®^, Vitulazio, Italy, at 450 nm = 40 mW/cm^2^). Stock solutions of curcumin (**1**) and curcumin-based plasticizers (**3** and **4**) were prepared in DMSO followed by preparation of solutions in H_2_O. Then, the solutions of curcumin (**1**) and curcumin-based plasticizers (**3** and **4**) in H_2_O were added to a 24-well plate and diluted using H_2_O at a concentration of 30 µM. The solutions of curcumin (**1**) and curcumin-based plasticizers (**3** and **4**) were illuminated, and aliquots (2 mL) were taken at specific light doses (0, 9.4, and 23.5 J/cm^2^) and analyzed using UV–Vis spectroscopy (200–800 nm). The photodegradation (%) was obtained by analysis of the decrease in the maximum absorbance peak at 430 nm (for curcumin), 390 nm (curcumin derivative **3**) and 424 nm (curcumin derivative **4**). Three experiments (n = 3) were carried out and the results were obtained and reported as means (± SD deviation).

### 3.6. Preparation of PVC–Curcumin-Based Films

The PVC-based films were made by using an Elcometer Film Applicator on a 20 × 20 cm glass surface. A 100 mL beaker with THF (9 mL) was heated to 40 °C on a stirring hotplate. Then, 500 mg of phthalate-free PVC was added and left to stir for 40–45 min. When the solution became viscous, a portion was placed on the doctor blade instrument, producing the film on a glass plate (20 cm × 20 cm). Then, selected amounts of the photosensitive curcumin-based plasticizers **2–5** were added to the THF solution and films with different *w*/*w* curcumin derivative loadings on PVC (0.1%, 15% and 30% *w*/*w*) were prepared using the procedure described above. Films 19 cm long, 1.5 cm wide and ~0.025 mm thick were obtained (Figure 3).

### 3.7. Characterization of PVC–Curcumin-Based Films

UV-Vis spectra of the solid materials were obtained on a Cary 5000 UV-Vis-NIR, where small pieces of the film were cut and placed in an adapter for solids and the scanning was performed from 200 to 800 nm. Differential Scanning Calorimetry (DSC) experiments were performed using a Perkin Elmer Pyris 1 power compensation calorimeter (Perkin Elmer, Waltham, MA, USA), equipped with a Cryofill cooling unit and a 20 mL/min helium purge. Samples (4 to 5 mg) were placed in Perkin-Elmer 30 μL aluminum pans. An identical but empty pan was used as reference. The samples were heated from 0 °C to 180 °C at 50 °C/min. Thermogravimetric analysis (TGA) was measured in a Perkin Elmer Simultaneous Thermal Analyzer, (STA) 6000. The study was carried out with heating up to 800 °C, at a heating rate of 10 °C/min under 20 mL/min nitrogen purge. Mechanical tensile strength tests of the films were performed in Hegewald & Peschke-Inspekt soil equipment (Hegewald & Peschke, Nossen, Germany); the load cell used was 500 N and the test speed was 5 mm/min by measuring the force required to break the material and the extent to which that material has the ability to stretch to its breaking point [44,45]. The mechanical stress is the maximum load that a material can withstand without fracture when stretched, divided by the original cross-sectional area of the material (Nm^−2^ or Pa). The samples (with average 25 µm) were conditioned for 48 h at a temperature of 23 ± 2 °C and the test temperature was 24 °C. Tensile strength and elongation at break were obtained from the stress–strain data. Each test used 3 replicates.

### 3.8. Photoantibacterial Evaluation of Photosensitive PVC–Curcumin-Based Films

For the photoinactivation studies, *Staphylococcus aureus* ATCC 29213 was used as model. From fresh overnight cultures in Mueller Hinton (MH) agar, an aqueous suspension was prepared with a density corresponding to 0.5 in the MacFarland scale, which is equivalent to 1−2 × 10^8^ CFU/mL. In flat-bottom 96-well plates, each well was covered with a circular section of each PVC–curcumin film (0.7 cm diameter). Then, 20 μL of the bacterial inoculum were added and the wells were irradiated with a blue LED lamp (415 nm, 6 mW/cm^2^), with a total light dose of 9.4 to 23.5 J/cm^2^. For the dark controls, the 96-well plates were covered with aluminum foil during the irradiation time. After irradiation, 80 µL of dH_2_O were added to each well and 10 μL aliquots were taken, diluted and plated in Petri dishes containing MH agar. Then, after a 24 h incubation time at 37 °C, the colony-forming units (CFU) were counted. Data statistical analysis was carried out in GraphPad Prism 8.0 (GraphPad Software, San Diego, CA, USA) using paired Student’s *t*-test. Statistical significance was assessed under *p* < 0.001 confidence interval.

### 3.9. Cytotoxicity Evaluation of Curcumin-Based Plasticizer **4**

The human dermal fibroblast cells (HDFn—ATCC, Germany) were grown by adhesion using the following supplemented medium: Dulbecco’s Modified Eagle’s Medium (DMEM) with antibiotics and 10% (*v*/*v*) fetal bovine serum (FBS). Following a standard procedure, the cells were preserved at 37 °C and in an atmosphere of CO_2_ (5%) and the cells were cultured with trypsin (0.25%) and EDTA (0.53 mM). Regarding the cell viability analysis, the standard MTT assay (3-(4,5-dimethylthiazol-2yl) -2,5-diphenyltetrazolium bromide), which is based on metabolic activity, was applied. Cells were seeded in complete medium in 96-well plates using a density of 1 × 10^4^ cells/well and incubated for c.a. 14 h (overnight). Then, medium was substituted by the sample solutions (curcumin derivative **4**) at 2.5, 5 and 10 µg/mL and DMEM (w/o phenol red, 5% FBS). For each sample concentration, a control was performed with the equivalent solvent concentration. After 24 h, the solutions were removed and cells were incubated using a solution of MTT (0.5 mg/mL) in DMEM (w/o phenol red, 5% FBS) for 3 h. Then, the formed formazan crystals were diluted using DMSO and the absorbance was analyzed on a plate reader (Multiskan™ FC Microplate Photometer) at 570 nm (corrected by at 690 nm). The results are expressed as mean ± standard deviation relative to control (cells incubated with the same solvent concentration). Each condition was carried out for all experiments and three independent experiments were conducted (n = 3). Concerning the statistical studies, these were carried out using Origin software (version 2018, Originlab, Northampton, MA, USA). For that, one-way ANOVA and the Tukey test were applied (to estimate the differences). To analyze the results of rejecting normality, the Kruskal–Wallis ANOVA approach was applied. *p* < 0.05 was applied to analyze the results that are statistically significant.

## 4. Conclusions

These studies clearly put in evidence that the size and structure of ester and ether curcumin derivatives have very significant effects on the modulation of plasticizing and antimicrobial properties of PVC–curcumin-based materials. Among the curcumin ester and ether herein synthesized (**1**–**4**), the PVC(**4**)-etherC10 film, with a 30% *w*/*w* curcumin derivative loading, showed the best mechanical properties to be used as bioplasticizer, corroborated by a decrease of Tg values by 18 °C, a decrease in the tensile strength by 8 MPa, along with a clear increase in elongation at breaking tension of 11%, with respect to PVC. Additionally, this PVC(**4**)-etherC10 material also showed a significant light-dose-dependent antibacterial photoinactivation efficiency against *S. aureus* planktonic cultures, displaying > 6 log CFU reduction at 23.5 J/cm^2^ light dose, concomitantly with no cytotoxicity against human fibroblast cells. 

In sum, this structure–activity study opens the way for future development of non-toxic photosensitive antimicrobial bio-based PVC–curcumin polymers for biomedical applications. This material may be considered an excellent alternative for replacing the toxic phthalate-PVC medical devices currently in clinical use.

## Data Availability

The data presented in this paper are available in the Appendix A.

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
