# Peer review of "Photoantibacterial Poly(vinyl)chloride Films Applying Curcumin Derivatives as Bio-Based Plasticizers and Photosensitizers"

_molecules, 2023, doi:10.3390/molecules28052209_

Round 1
Reviewer 1 Report
Manuscript tittled: Photoantibacterial poly(vinyl)chloride films applying curcumin derivatives as bio-based plasticizers and photosensitizers
This manuscript is about a modified PVC film endowed with antimicrobial features. Information presented in this manuscript is appropriate for Molecules. Some queries and misspells found as follows
Line 32, change to “…building and construction…”
In Figure 2 line 147, what is the light source?
Line 154. Could you explain with a scheme the “doctor blade type film coating procedure”
Figure 6a) and 6b).Y-axis units are MPa, check capitals in P
Figure 6b) Y-axis 10MPa is missing
Line 545. Check °C symbol
In References section, check the style to cite since many inconsistencies are recognizable.
Author Response
Reviewer #1
This manuscript is about a modified PVC film endowed with antimicrobial features. Information presented in this manuscript is appropriate for Molecules. Some queries and misspells found as follows
Line 32, change to “…building and construction…”
In Figure 2 line 147, what is the light source?
Line 154. Could you explain with a scheme the “doctor blade type film coating procedure”
Figure 6a) and 6b).Y-axis units are MPa, check capitals in P
Figure 6b) Y-axis 10MPa is missing
Line 545. Check °C symbol
In References section, check the style to cite since many inconsistencies are recognizable.
Answer: we thank the reviewer for the manuscript analysis and review. The misspells and queries were all corrected. A figure at the supporting information was included, illustrating the doctor blade type film coating procedure (Figure S22).
Reviewer 2 Report
1. I suggest authors to collect ad include IR spectroscopic data of the reported compounds.
2. It is quite surprising to see no degradation of curcuminoids in TGA. Authors should explain this.
3. Discuss the effect of structural changes on absorption/emission spectra.
Author Response
Reviewer #2
Comments and Suggestions for Authors
- I suggest authors to collect ad include IR spectroscopic data of the reported compounds.
Answer: We thank the reviewer for the manuscript review analysis. We have included the IR spectroscopic data in experimental section and corresponding figures in Supporting Information.
- It is quite surprising to see no degradation of curcuminoids in TGA. Authors should explain this.
Answer: We have observed, by TG and dTGA (see Figure S19 in Supporting Information), that the thermal profile of e.g. curcumin (1) indicates degradation at temperatures between 250 ºC and 450 ºC. However, this event is indistinguishable when performing the thermal decomposition of the PVC-curcuminoid films, since both thermal events (PVC and curcuminoid) occur simultaneously. The text was changed to clarify this issue.
“This phenomenon occurs similarly for the PVC-curc film (see Figure S19), in which the addition of curcumin to PVC does not affect the thermal decomposition profile, when compared to PVC (black), which demonstrates the high thermal stability of the doped PVC-curc material. The same thermal behavior was observed for all the other PVC-curcuminoid films.”
- Discuss the effect of structural changes on absorption spectra.
Answer: Sentences were added below figure 1, discussing the effect of structure in the absorption spectra of the synthesized compounds.
“C10-di ether derivative 4 shows a λmax = 422 nm (Figure 1, red line), which is almost coincident with the typical curcumin (1) spectrum (Figure 1, black line) [27]. This indicates that the presence of electron-donating ether groups does not significantly change the electron conjugation. Contrarily, the sub-product C10-triether derivative 5, displays a λmax = 380 nm. This large ipsochromic shift may be attributed to the reduction of conjugation across the molecule due to inhibition of the keto-enol equilibrium. Regarding the UV-Vis absorption spectra of di-ester curcumin derivatives 2 and 3, shorter ipsochromic shifts are observed, λmax= 400 nm (Figure 1 violet, blue and yellow lines), when compared with 5 [28,29]. This occurrence may be attributed to the reduction of conjugation due to the transformation of curcumin hydroxyl groups to the correspondent electron withdrawing ester groups.”